# Weather Volatility and Production Efficiency

**Denitsa Angelova** [1,*] and **Jan Käbel** [2]

1   School of Management, Technical University of Munich, Arcisstraße 21, 80333 Munich, Germany
2   School of Life Sciences Weihenstephan, Technical University of Munich, Alte Akademie 8,
    85354 Freising, Germany; jan.kaebel@tum.de
*   Correspondence: denitsa.angelova@tum.de

**Abstract:** We formulate a stochastic production frontier model to estimate the production efficiency scores while correcting for technical progress and weather effects in the form of temperature and precipitation levels and volatility. We econometrically estimate a model for European agriculture. Our results indicate that average temperature, unlike average precipitation levels, significantly influences aggregate agricultural output. We estimate that a marginal increase in temperature would decrease aggregate European agricultural output by about 1.6% percent. Further estimation results indicate a slight increase in output associated with marginal increases of precipitation and temperature volatilities.

**Keywords:** production efficiency; agriculture; weather

## 1. Introduction

Changes in climate, i.e., the average weather patterns over longer periods, constitute a significant, tangible, and costly challenge to humanity in the twenty-first century [1]. The phenomenon is defined by the Intergovernmental Panel on Climate Change (IPCC) as "a change in the state of the climate that can be identified (e.g., by using statistical tests) by changes in the mean and/or the variability of its properties and that persists for an extended period, typically decades or longer" [2].

The process behind the climatic changes seems deceptively well understood. The earth's system warms up when solar radiations pass through the earth's atmosphere. These radiations also warm up greenhouse gasses, which reradiate lower wavelengths that would be reflected back into space under normal circumstances. This process causes the sea and land surface temperatures to rise, accompanying a change in sea levels, storm intensity, snow caps and glaciers, warmer air, increased humidity, and changes in in rainfall frequency [3]. Human activities contribute to this process by increasing the greenhouse gasses concentration in the atmosphere [4,5]. IPCC has declared current climatic developments to be "unequivocal" and "unprecedented", and established the last three decades as the warmest since 1850 [2]. The frequency of heat waves and of heavy precipitation has likely increased in large parts of Europe since 1950 [5].

Climate scientists expect surface temperature to increase in the range of 0.3–0.7 °C by the year 2035 relative to the period 1986–2005 [2]. Forecasts beyond this point are challenging due to the feedback loops between the climate and economic systems and are systematically summarized in so-called representative concentration paths (RCP) named after the forecasts of radiative forcing values they entail for the year 2100. RCP 2.6, RCP 4.5, RCP 6.0, and RCP 8.5 in W/m$^2$ are typically closely examined [5].

Europe until the year 2035 is likely to experience more warm and hot days and nights, while the number of cold days and nights will decrease. The number of hot days and nights is also likely to increase. For the late 21st century, it is very likely that more heat waves will occur over most land areas

and that extreme weather events will occur more frequently and more severely. There is also a greater risk of longer and more severe droughts. In addition, it is very likely that the sea level will reach a higher level in the late 21st century [5].

Agriculture has a direct connection with the climate [6]. Due to this direct interaction with the environment, the agricultural sector is particularly dependent on the effects of changing conditions. Two important features of the natural environment regarding agricultural production, temperature, and water availability, are likely to be defined by frequent shocks as a result of climatic changes [7,8]. Incoming radiation also affects the process of photosynthesis, which is the basis for crop production [3]. Moreover, temperature has a direct impact on the duration of the growth period for the crop production [9]. Extreme weather events also matter. In the case of crop production, for instance, they can directly physically damage the crops or affect plant physiological processes such as photosynthesis and evapotranspiration [10].

Due to the uncertainty of the future climatic and economic development, it is difficult to make an exact forecast of the consequences in detail [11]. In terms of food security and in the light of the recent wave of observed temperature extremes, it is nevertheless important to attempt a quantification of climatic effects on the agriculture as these effects are inevitably transmitted to other sectors of the world economy along the value chain.

This study contributes to the debate by providing a methodology for the quantification of climatic effects by formulating a stochastic frontier model, which frames weather effects, temperature and precipitation levels and volatilities, as factors influencing agricultural production efficiency. This allows for the estimation of the sensitivity of aggregate agricultural output to marginal changes in weather conditions. We hypothesize that all environmental variables introduced in the analysis significantly affect the aggregate output and test this hypothesis statistically.

The structure is as follows: Section 2 provides a preview of existing studies linking productivity and efficiency to weather and climate effects, Section 3 describes the methodical framework and the corresponding empirical specification, Section 4 describes the data, Section 5 presents the estimation results and a discussion, and Section 6 concludes.

## 2. Literature Review

Several recent contributions address the issue of weather and climate effects on productivity and efficiency with total factor productivity defined as the ratio of aggregate output to aggregate input and used as a measure of efficiency. Letta and Tol (2016) investigate the effects of temperature and precipitation shocks on total factor productivity from a macroeconomic perspective by examining the equation describing the production technology found in integrated assessment models like DICE (the Dynamic Integrated Climate-Economy model) [12]. More specifically, they introduce a factor explicitly accounting for the hypothesized dependence of the technical change on weather shocks, which are presumably unanticipated by economic agents, and subsequently econometrically test for it. The preliminary estimation results shows the significance of weather shocks in the case of less developed economies. Their analysis is, however, somewhat limited by the fact that the authors test their hypothesis using data on an aggregated national level. Since different sectors of the economy cannot necessarily be assumed affected by weather shocks in the same magnitude, it would be preferable to use sector level data to safeguard for sectoral specific effects balancing each other out and obscuring the relationships of interest.

A study dedicated to the investigation of climate effects on total factor productivity in US agriculture is Liang et al. (2017) [13]. The study describes the correlations found between regional climate patterns and national level of total factor productivity (TFP) measures. They find that temperature or precipitation explain 70 percent of total factor productivity variation. The growth in total factor productivity for the US case will nevertheless remain positive, as the efficiency increases due to innovation activity in US agriculture are large enough to offset the negative effects of changing

weather patterns. The growth in total factor productivity due to weather changes will be curbed by some 1.5 percent by the mid-century if the relationship continues.

Njuki, Bravo-Ureta, and O'Donnell (2018) decompose the growth of a measure of total factor productivity derived from a formulation of the production technology into weather effects, technological change, changes in technical efficiency and changes in scale-mix efficiency [14]. The estimation uses US state-level data for the period between 1960 and 2004 and accounts for growing season temperature and precipitation, as well as intra-annual standard deviations of the growing season temperature and precipitation. The results indicate an annual growth of total factor productivity of 1.5 percent and a small, but statistically significant decline of total factor productivity growth by 0.012% due to weather effects.

Wang et al. (2018) use a stochastic frontier production framework to estimate the impact of extreme events on US agricultural productivity based on historical weather data from the period between 1940 and 1970 [15]. They estimate efficiency while accounting for two compound temperature-humidity indices, namely the Temperature Humidity Index (THI) and the Oury index (1965), to capture the weather and climate effects [16]. They find that a higher THI index, which is consistent with more heat waves, and a lower Oury index, which is consistent with drier conditions, contribute to a lower productivity. Projections based on the estimation results indicate an uneven distribution of climate effects on agricultural productivity, with the Northeast, Southeast, and Mississippi Delta being the most highly affected areas.

Zhong, Hu, and Jiang (2019) investigate the climatic and weather determinants of Chinese province level total factor productivity [17]. In a first step, they use a deterministic data envelopment approach to calculate the total factor productivity as a ratio between inputs and outputs. They subsequently employ a spatial econometric model explaining total factor productivity as a function of state investment in agricultural research and development and a range of weather variables such as total annual precipitation, average growing season temperature, and intensity of evapotranspiration. The effects of weather variables on the so-defined total factor productivity are negative and statistically significant.

## 3. Model and Empirical Specification

This study uses the simplest possible formulation of the production technology, which allows for efficiency considerations, namely a single-output, single-input formulation. The total input $x$ committed to production results in the total output $y$ so that the radial expansion characteristic of the output distance formulation of the production technology with the help of the auxiliary variable $\theta$ holds at:

$$D_0(x, y) = \min\left\{\theta : \frac{y}{\theta} \in Y(X)\right\}. \tag{1}$$

Individual countries could produce beneath the technologically feasible frontier described by the output distance formulation above. This study hypothesizes that this could be the result of weather conditions and defines a truncated normal stochastic term $u_{it}$ containing these weather conditions for a country $i$ and period $t$ as formalized in (2).

$$y_{it} = e^{\beta_0 u_{it}} x_{it}^{\gamma} \tag{2}$$

This term $u_{it}$ is a function of weather variables $z_{it}$ and neutral technological progress $t_{it}$ as described in (3).

$$u_{it} = \delta_0 + \varphi t_{it} + \sum_{z=1}^{Z} \delta_z z_{it} \tag{3}$$

Following Battese and Coelli (1995) the stochastic deviations from the technologically feasible are captured in the log-linearized empirical specification derived from (2) by introducing the truncated normal stochastic term $u_{it}$ in Equation (4) [18]. The formulation of the production technology spans

over the two Equations (4) and (5). It involves a single aggregate input $x_{it}$ and a single aggregate output $y_{it}$ as well as the stochastic error terms $v_{it}$ and $w_{it}$.

$$ln(y_{it}) = \beta_0 + \gamma ln(x_{it}) + v_{it} + u_{it} \tag{4}$$

$$u_{it} = \delta_0 + \sum_{z=1}^{Z} \delta_z z_{it} + \varphi t_{it} + w_{it} \tag{5}$$

## 4. Data

This contribution relies on merging economic and weather information. Economic data originates from the Farm Accountancy Data Network (FADN), which was established by the EU Commission in 1965. The aim of the FADN is to collect data on the accounting, income and business of agricultural holdings in the EU. The data collected serve as a basis for regular reports on the state of agricultural businesses and market [19]. The collection of data and the associated monitoring function represents an essential element in the development of CAP. The data collection of the FADN takes place based on a sample of selected farms from all 28 member states, which represent large parts of the cultivated area and the agricultural production of the EU. The data are collected at the local level by 143 divisions subdivided geographically by Member States [20].

The FADN variables used are SE131-Total output and SE270-Total Inputs, both measured in billions of Euro. Total Output includes all crops, crop products, livestock, livestock products, and other output. Changes in stocks and valuation are reflected. Total inputs includes all costs linked to the agricultural activity including specific and unspecific costs as well as on-farm use of own products [21]. The dataset in use consists of aggregated farm data at the regional and country level and covers all 28 EU member states. Besides their geographical origin, agricultural holdings are also grouped in six size classes by their gross production. For 25 out of all 28 EU member states, data are available for 14 consecutive years from 2004 to 2016. As Bulgaria and Romania only entered the EU in 2007 and Croatia entered the EU in 2013, data for those countries are provided for the latest eleven or five years. In total, 270 values across 21 member states (Austria, Belgium, Bulgaria, Denmark, Estonia, Finland, France, Germany, Greece, Hungary, Ireland, Italy, Latvia, Luxembourg, Netherlands, Poland, Portugal, Slovakia, Slovenia, Spain, and the United Kingdom) and 13 years were used in the analysis due to data concerns in the corresponding weather observations.

Data on weather developments in Europe originate from the European program for earth observation known as Copernicus. In 1998, the European Commission decided to establish a Global Monitoring for Environment and Security (GMES), which would enable joint monitoring of the environment and security by means of satellites. The aim was also to establish a global information service that would provide an interface between collected data and citizens and institutions. In 2012, the European Commission renamed GMES into Copernicus. The opportunities resulting from the program focus on six thematic aspects: atmosphere, sea, land, climate change, security and emergencies [22].

The county level weather data were extracted from the data set for the European energy sector from the Copernicus program called European Climatic Energy Mixes (ECEM). The underlying data are from the European Reanalysis Interim (ERA) provided by the European Center for Medium Range Weather Forecasts (ECMWF) on six hour basis. The data set consists of eight different weather variables for the period from 1979 to 2016 with an aggregated temporal resolution of one day, month, or year. The spatial resolution is a grid of $0.5° \times 0.5°$. Data is available for clusters from the e-Highway2050 project and aggregated on country level [23]. For further investigation, air temperature and precipitation were selected as the variables. Air temperature describes the temperature at a height of 2 m above the surface [24], and precipitation describes the accumulated height of rainfall on a flat surface within a certain period of time [25]. In total, temperature and precipitation data were available for 26 of 28 members of the EU on a daily basis. We used the data on temperature and precipitation to construct variables indicating the national annual mean and the variation within a year on a national level.

We cleaned and matched the economic and weather data for the European countries depicted on Figure 1. The descriptive statistics of the dataset are reported in Table 1.

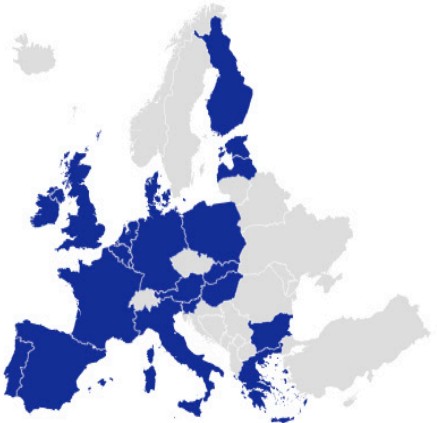

**Figure 1.** Location of the European countries that the data originated from. Source: Own illustration.

In Table 1, we provide information on the mean, maximum, and minimum values for the variables. The values for the first and third quartile are reported to allow an inference on the shape of the distribution of the data, as fifty percent of the observations for individual variables lie between the two. While the economic data exhibits a certain regularity, we can observe that the distributions of the weather variables are skewed. The minimum observed annual temperature of 0.880 and the highest maximum temperature variance of 161.09 can be traced back to Finnish weather conditions in 2010, which were characterized by a cold winter and an untypically warm summer featuring a record national high of 37.2 °C [26]. The unusually large amount of precipitation implied by the value 2269.0 mm was measured in Slovenia in the year 2014 as the country experienced a wave of severe and long-lasting rain [27]. Both extreme inter-annual variations in precipitation, 3.701 and 102.452, respectively, stem from the year 2005 when the Southeast of Spain was exhibiting severe drought while Slovenia and a range of countries in Central and Eastern Europe were facing floods [28,29].

**Table 1.** Summary Statistics of the Variables: Mean, Minimum, Maximum, Lower (1st) Quartile, Upper (3rd) Quartile.

| Variable | Mean | Min | Max | 1st Qua | 3rd Qua |
|----------|------|-----|-----|---------|---------|
| Total Output | 16.00 | 14.09 | 18.23 | 15.21 | 16.88 |
| Total Input | 11.335 | 9.394 | 13.626 | 10.374 | 12.233 |
| Temperature °C | 10.232 | 0.880 | 16.490 | 8.652 | 12.182 |
| Precipitation mm | 896.4 | 393.9 | 2269.0 | 706.8 | 996.2 |
| Temperature Variance | 55.79 | 13.30 | 161.09 | 39.88 | 70.17 |
| Precipitation Variance | 17.003 | 3.701 | 102.452 | 8.667 | 17.244 |

Source: Own calculation based on FADN and Copernicus Climate Data Service.

## 5. Results and Discussion

We apply panel data tests for the presence of unit roots in order to assess the necessity of applying nonstationary data regression methods such as the method developed by Pesaran et al. (1999), Granger and Hyung (1999), or Binder, Hsiao, and Pesaran (2005) [30–32]. Unit root tests in general would uncover the presence of a stochastic trend in the data. We reject the null hypothesis of a unit root with a variety of tests as reported in Table 2 in favor of stationarity or the lack of a stochastic trend based on standard significance levels and regard the common statistical tests as reliable in this case. Table 2 displays the results of the unit root tests obtained for a maximum of 4 lags and allowing for an individual intercept and trend with the purtest function in the R package plm.

**Table 2.** Panel unit root tests performed and the associated p values.

| Test | $p$ **Value** |
|---|---|
| Maddala-Wu Unit-Root Test | $1.677 \times 10^{-9}$ |
| Levin-Lin-Chu Unit-Root Test | $9.196 \times 10^{-8}$ |
| Im-Pesaran-Shin Unit-Root Test | $5.968 \times 10^{-11}$ |
| Choi's modified P Unit-Root Test | $2.2 \times 10^{-16}$ |
| Choi's Inverse Normal Unit-Root Test | $6.662 \times 10^{-11}$ |
| Choi's Logit Unit-Root Test | $3.288 \times 10^{-9}$ |

Source: Own calculation based on the data described in Table 1.

We estimate the empirical specification (2) via maximum likelihood using the R package frontier. Table 3 displays the estimation results. They indicate that European agriculture exhibits decreasing returns to scale, as indicated by the statistically significant coefficient associated with the total input of the size of approximately 0.91. This means that increasing the aggregate input would increase aggregate output by a factor of less than one. The results are significant at the 1% level, as the t statistic, the p values and the significance code indicate.

**Table 3.** Regression results.

| Variable | Coefficient | Std. Error | Pr(>\|z\|) | Significance | Log Likelihood | $\delta_z^2$ |
|---|---|---|---|---|---|---|
| Intercept | 5.7973 | 0.085851 | 0.000 | *** | 165.5608 | 0.017176 (***) |
| Total Input | 0.90844 | 0.0074697 | 0.000 | *** | | |
| Temperature (Z) | −0.015298 | 0.0025424 | 0.000 | *** | | Mean efficiency |
| Precipitation (Z) | −0.000046781 | 0.000047978 | 0.3295298 | | | 0.9177793 |
| Temperature Var (Z) | 0.0039076 | 0.00033215 | 0.000 | *** | | $\gamma$ |
| Precipitation Var (Z) | 0.0032452 | 0.0008.5311 | 0.000 | *** | | 0.0001.5378 (***) |

Source: Own estimation based on the data described in Table 1. Significance codes: *** 0; ** 0.001; * 0.01.

As reported in Table 3, the mean estimated efficiency in the model is around 92%. We display the individual country's annual efficiency estimates in Table 4 and in Figure 2. Efficiency levels close to unity indicate that the production in the specific country takes place on the production frontier or, in other words, that the country is as efficient as it could be. Table 3 illustrates the annual efficiencies for each country and the overall mean, which we calculate as the sum of annual efficiency score over the entire span of the observations divided by the length of the observation period.

**Table 4.** Rounded up estimated efficiencies.

| | 04 | 05 | 06 | 07 | 08 | 09 | 10 | 11 | 12 | 13 | 14 | 15 | 16 |
|---|---|---|---|---|---|---|---|---|---|---|---|---|---|
| **Austria** | 0.8798 | 0.8321 | 0.8366 | 0.9043 | 0.9026 | 0.8465 | 0.8226 | 0.8805 | 0.8359 | 0.8598 | 0.966 | 0.8984 | 0.9022 |
| **Belgium** | 0.9969 | 0.987 | 0.9652 | 0.9999 | 0.9998 | 0.9785 | 0.9031 | 0.9999 | 0.9905 | 0.9595 | 0.992 | 0.9999 | 0.9936 |
| **Bulgaria** | NA | NA | NA | 0.8976 | 0.903 | 0.916 | 0.8823 | 0.8664 | 0.7915 | 0.9171 | 0.9563 | 0.9061 | 0.8871 |
| **Denmark** | 0.9811 | 0.9724 | 0.9275 | 0.9999 | 0.9996 | 0.9625 | 0.8449 | 0.972 | 0.9551 | 0.9338 | 0.9996 | 0.9999 | 0.971 |
| **Estonia** | 0.8275 | 0.7844 | 0.7499 | 0.8217 | 0.9276 | 0.8145 | 0.6395 | 0.7701 | 0.7432 | 0.7826 | 0.8092 | 0.9455 | 0.8046 |
| **Finland** | 0.7206 | 0.7161 | 0.6575 | 0.7157 | 0.8413 | 0.7049 | 0.5469 | 0.6746 | 0.6624 | 0.7019 | 0.7276 | 0.8345 | 0.6956 |
| **France** | 0.9999 | 0.9839 | 0.998 | 1 | 1 | 0.9998 | 0.9704 | 1 | 0.9998 | 0.9998 | 1 | 0.9999 | 0.9999 |
| **Germany** | 0.9632 | 0.9375 | 0.916 | 0.9965 | 0.9917 | 0.9425 | 0.8646 | 0.9779 | 0.936 | 0.923 | 0.9999 | 0.9916 | 0.965 |
| **Greece** | 0.9788 | 0.9574 | 0.9388 | 0.9738 | 0.9854 | 0.9997 | 0.9982 | 0.9578 | 0.908 | 0.9995 | 0.9999 | 0.9577 | 0.9681 |
| **Hungary** | 0.8922 | 0.8338 | 0.859 | 0.9205 | 0.9322 | 0.8866 | 0.8478 | 0.8648 | 0.8226 | 0.8949 | 0.9771 | 0.9039 | 0.8867 |
| **Ireland** | 1 | 1 | 0.9999 | 1 | 1 | 0.9999 | 0.9864 | 1 | 1 | 0.9999 | 1 | 1 | 1 |
| **Italy** | 0.9999 | 0.9738 | 0.9995 | 0.9999 | 0.9999 | 0.9972 | 0.9856 | 0.9999 | 0.9659 | 0.9999 | 1 | 0.9999 | 1 |
| **Latvia** | 0.8402 | 0.7966 | 0.7636 | 0.8375 | 0.9392 | 0.8399 | 0.671 | 0.7993 | 0.7599 | 0.7855 | 0.8183 | 0.9409 | 0.8267 |
| **Luxembourg** | 0.9482 | 0.927 | 0.9217 | 0.9898 | 0.973 | 0.9239 | 0.8654 | 0.994 | 0.9444 | 0.9047 | 0.9999 | 0.9634 | 0.9505 |
| **Netherlands** | 0.9995 | 0.9997 | 0.9703 | 0.9999 | 0.9998 | 0.9861 | 0.901 | 0.9999 | 0.9896 | 0.9617 | 1 | 0.9999 | 0.9997 |
| **Poland** | 0.9101 | 0.8805 | 0.819 | 0.9354 | 0.9661 | 0.8908 | 0.778 | 0.8875 | 0.827 | 0.8674 | 0.9261 | 0.9525 | 0.9007 |
| **Portugal** | 1 | 1 | 0.9999 | 1 | 1 | 1 | 0.9999 | 1 | 1 | 0.9999 | 1 | 1 | 1 |
| **Slovakia** | 0.8685 | 0.8138 | 0.8196 | 0.8947 | 0.9149 | 0.8587 | 0.7869 | 0.8628 | 0.7995 | 0.8615 | 0.9469 | 0.8917 | 0.8651 |
| **Slovenia** | 0.749 | 0.6722 | 0.7082 | 0.7627 | 0.7927 | 0.7728 | 0.6949 | 0.7288 | 0.718 | 0.7588 | 0.832 | 0.7379 | 0.8195 |
| **Spain** | 0.9999 | 0.9992 | 0.9999 | 1 | 1 | 0.9999 | 0.9999 | 1 | 0.9998 | 0.9999 | 1 | 0.9999 | 1 |
| **United Kingdom** | 1 | 1 | 1 | 1 | 1 | 1 | 0.9934 | 1 | 1 | 0.9999 | 1 | 1 | 1 |

Source: Own estimation based on the data described in Table 1.

a) 2004

b) 2005

c) 2006

d) 2007

e) 2008

f) 2009

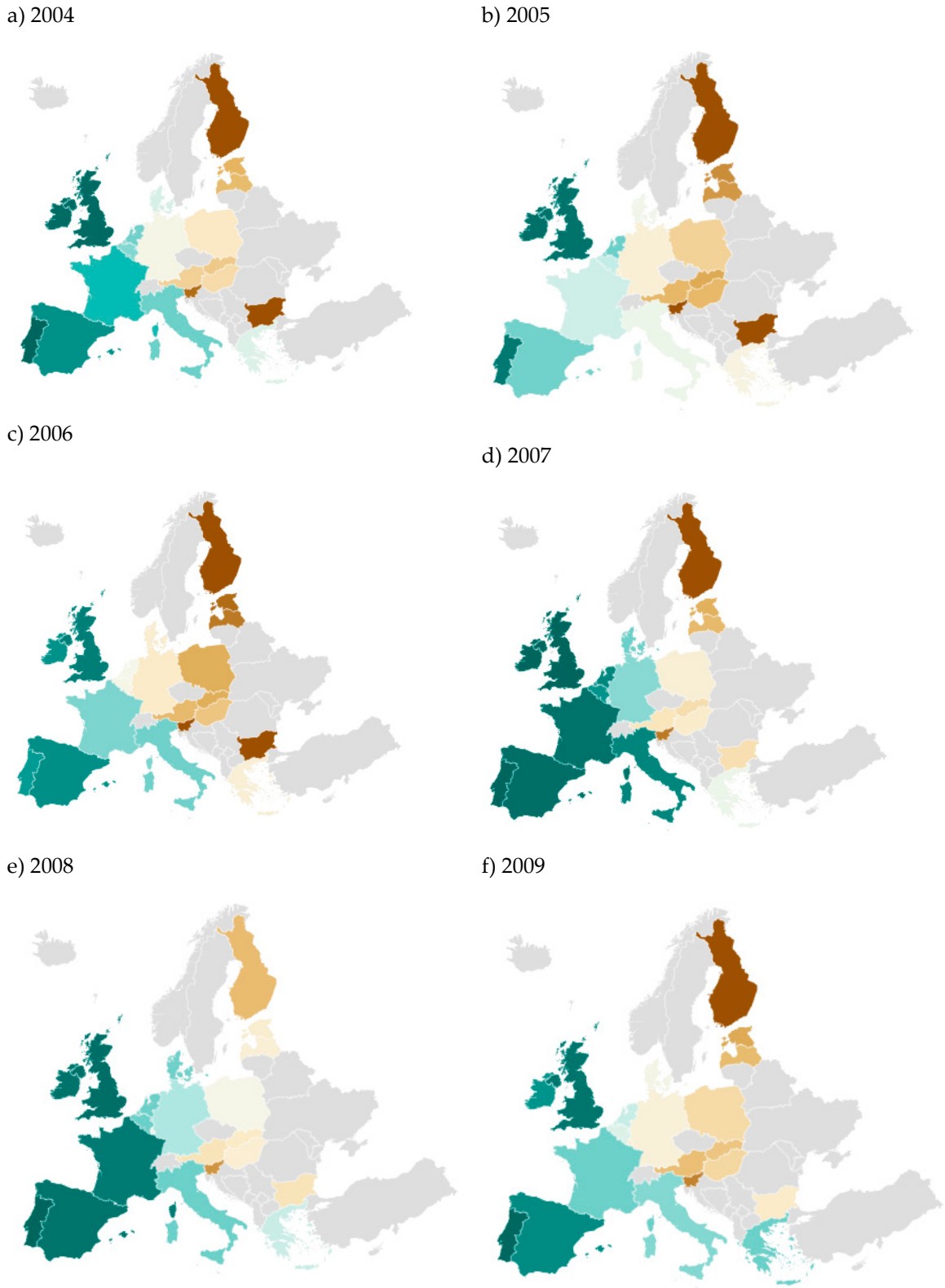

**Figure 2.** *Cont.*

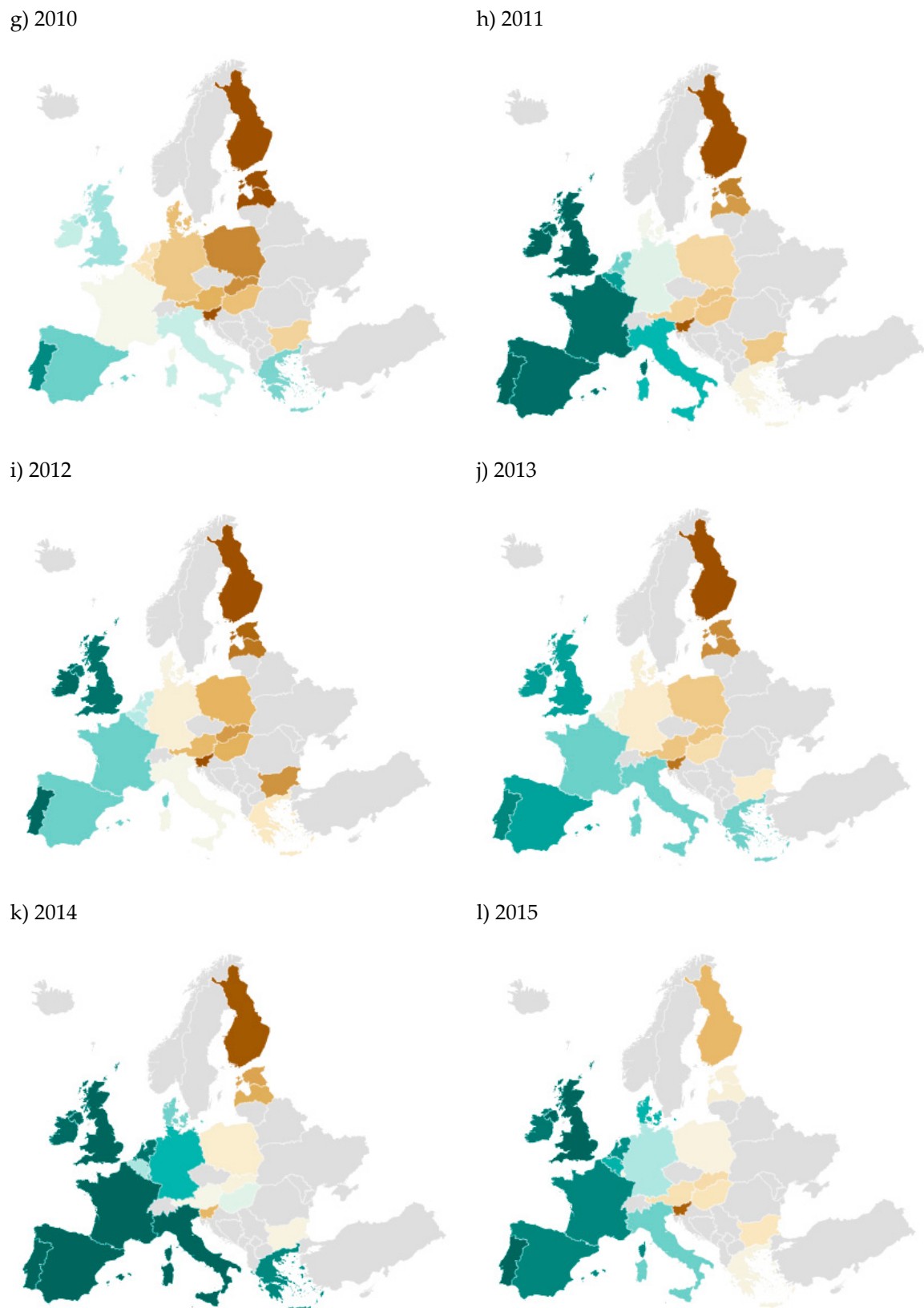

**Figure 2.** *Cont*.

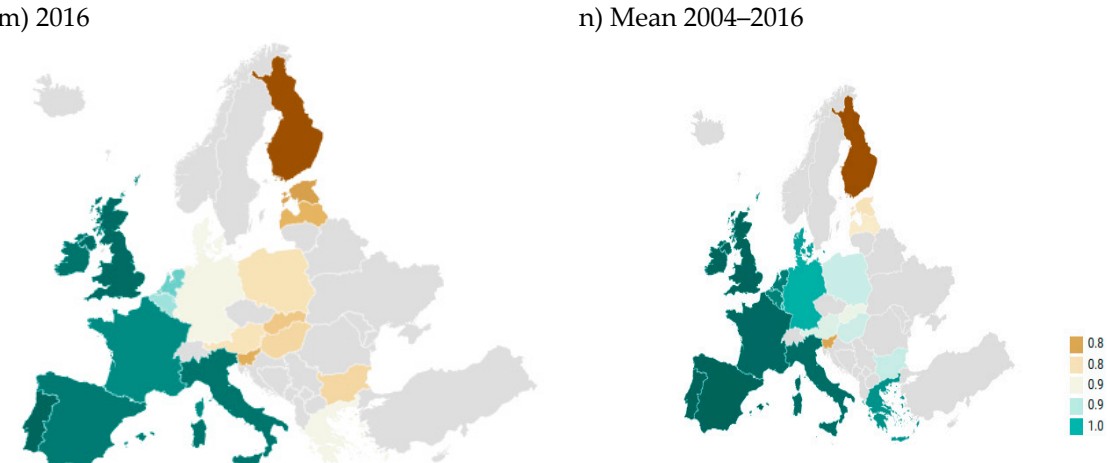

**Figure 2.** Estimated efficiencies for the period 2004–2016. Dark green indicates high efficiency, dark brown indicates low efficiency and pink indicates middle range values. Legend: Dark green indicates high efficiency, while dark brown indicates high inefficiency. Pinkish shades point to inefficiency scores in the middle range. Source: Own illustration based on the estimates in Table 4.

In general, we observe that Eastern and Northern European countries tend to exhibit lower levels of efficiency compared to Western European countries. These efficiency differences are partially explainable by the presence of former East Bloc production structures and conditions, as well as Finland, a country whose harsh and cold climate makes it generally unsuitable for agricultural production. As expected, the effects of climate extremes such as the 2010 European flood wave, the 2012 cold wave, and the year 2013, which brought a cold wave, flooding and a subsequent heatwave, are associated with discernably lowered efficiencies.

With respect to weather, we fail to find evidence that mean annual precipitation is a significant determinant of agricultural production in the European case. We, however, observe significance with respect to all the other environmental variables considered. An increase in mean temperature would lead to a ceteris paribus reduction in output as indicated by the negative sign of the statistically significant coefficient associated with the variable Temperature (Z). The size of the expected output reduction can be determined by calculating the marginal effect of an increase in temperature in Equation (2). As with the other inefficiency-affecting variables, a local evaluation of the marginal effect of a partial increase is necessary. An evaluation of the marginal product derived from Equation (2) at the sample mean indicates that a slight increase in the Temperature (Z) variable would lead to a reduction in output consistent with a loss of a quarter billion Euro or of about 1.56 percent of the average agricultural output. Likewise, a slight increase in temperature volatility captured by Temperature Var (Z) or precipitation volatility captured by Precipitation Var (Z) would both lead to an output increase and be consistent with gains of 50 and 60 million Euro on average, respectively, or of about 0.3 percent of average agricultural output. However, the economic factors to agricultural production captured by the variable Total Input remain the main determinant of agricultural output and contribute most to explaining the observed variation. The variance associated with the introduction of the weather variables $\delta_z^2$ explains a rather small fraction of the overall variance as indicated by the coefficient $\gamma$.

Given this small percentage of variance explained by the weather variables, it is reasonable to test the validity of the inefficiency model, which we do by the means of a likelihood ratio (LR) test. The loglikelihood of the unrestricted model with 8 degrees of freedom is 165.561, while the loglikelihood of the restricted model with 3 degrees of freedom is 91.447. After determining the value of the asymptotically $\times 2$ distributed likelihood ratio test statistic, we rejected the null hypothesis of the weather associated coefficients being jointly zero with a high degree of confidence given a *p* value of $2.2 \times 10^{-16}$. We concluded that the unrestricted model including the inefficiency terms offers a superior explanation of the observed data. In a Shapiro-Wilk setting, we failed to reject the null hypothesis of

normality of the regression residuals based on a p-value of 0.3191, which led us to believe that our simple linear model is an appropriate and valid simplification of reality.

## 6. Conclusions

We formulated a stochastic frontier production model capable of accommodating weather effects as a determinant of agricultural production inefficiency. We provided a way to calculate the expected marginal changes in output associated with an increase in the inefficiency determinants. We tested and confirmed the validity of our model statistically. Evidence of geographical heterogeneity is present in our study similarly to the results reported in the studies of Letta and Tol (2016), Liang et al. (2017), Njuki, Bravo-Ureta, O'Donnell (2018), Wang et al. (2018), and Zhong, Hu and Jiang (2019). We further find evidence of decreasing returns to scale similar to the studies of Njuki, Bravo-Ureta, O'Donnell (2018), and Wang et al. (2018), which focus on the US case.

Unlike many of the studies mentioned, we found no evidence that precipitation plays a significant role in the determination of aggregate agricultural output. This does not mean that it could not constitute a significant factor in the production environment in typically dry and warm areas such as the Iberian Peninsula or that this insignificance holds in general. Njuki, Bravo-Ureta, and O'Donnell (2018) found that an increase in precipitation would increase aggregate agricultural output for the US case and Wang et al. (2018) observed similar effects.

Temperature, on the other hand, plays a significant role in determining agricultural production inefficiency and thus affects aggregate agricultural output. An evaluation of the expectation of the marginal product at the sample mean indicates that a marginal increase in temperature would decrease total agricultural output by about 1.6% of the average agricultural output. A similar negative effect of a marginal temperature increase on aggregate agricultural output is reported for the cases of the US in Njuki, Bravo-Ureta, O'Donnell (2018), Wang et al. (2018), and for China in Zhong, Hu, and Jiang (2019).

We estimate that a marginal increase in either temperature and precipitation volatility would increase the aggregate agricultural output by roughly 0.3% to 0.4%. The only other study considering the second moment of the weather variables as a determinant of inefficiency, the study of Njuki, Bravo-Ureta, and O'Donnell (2018), finds in contrast that a marginal increase in precipitation volatility would decrease the US aggregate output and that temperature volatility is not statistically significant.

The analysis outlined provides pointwise estimates of inefficiency and sensitivity of aggregate output to marginal changes in weather conditions. An extrapolation of these estimation results would require that the estimated functional relationship between aggregate input and aggregate output holds in the future. This is a reasonable assumption if the production technology and farm practices remain the same. However, given the clearly negative impact of an increase in average temperature on aggregate output that will be difficult to offset by the estimated boost due to increased volatility, adaptation measures are called for. Practical measures for adaptation could include the adoption of drought resilient crop varieties or the adoption of water-saving irrigation technologies.

**Author Contributions:** D.A. did the conceptualization, the methodology and the formal analysis and was responsible for the project administration, the writing of the original draft and the subsequent review and editing of the manuscript. J.K. did the data curation and contributed to the literature investigation and to the writing of the original draft.

**Funding:** This work was supported by the German Research Foundation (DFG) and the Technical University of Munich (TUM) in the framework of the Open Access Publishing Program.

**Conflicts of Interest:** The authors declare no conflict of interest.

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
