# Peer review of "Weather Volatility and Production Efficiency"

_sustainability, doi:10.3390/su11246970_

Round 1
Reviewer 1 Report
This is an interesting topic. However, in general, the manuscript need to be improve.
General Comments: It is my impression that this manuscript is written for a very local audience in Europe. It could make sense, since its based on the European Union. However, too much specificity without the correspondent explanation, is difficult for international readers. Sometimes more description is needed in order for readers to keep focus.
I suggest the authors to internationalize the manuscript as much as possible in order to expand the potential audience and readers.
Some specific examples are such as: SE131; SE270 and many others local acronyms.
Specific Comments:
Abstract:
Line 13-15: Future work includes impact analysis via scenario development for temperature and precipitation similar to the work of Wang et al. (2018), yet correcting for the interdependencies between the two. Is it normal to include references in the abstract?? Please confirm this with Journal format and specifications.
Introduction: Climate and Agriculture is a discipline that existed even before the Agricultural Revolution. That said, there are many important references missing. Under the Climate Change context also mentioned in the manuscript, Food Security need to be considered in the discussion even thought I recognized it is not focus of your analysis.
Page 2. Line 47. Ibid? please confirm this citation format with Journal rules.
3. Model and Empirical Specification. Equations are numbered, however there is no reference in the text. Also, equation numbers are out of order.
Please add a conclusion Section.
References are not following any specific format. Please confirm with Journal format. Authors Last name are cited on the text, however, in the reference section, "References" are numbered. This makes the reviewer's job difficult to verify the correct use of scientific literature.
Author Response
This is an interesting topic. However, in general, the manuscript need to be improve.
General Comments: It is my impression that this manuscript is written for a very local audience in Europe. It could make sense, since its based on the European Union. However, too much specificity without the correspondent explanation, is difficult for international readers. Sometimes more description is needed in order for readers to keep focus.
I suggest the authors to internationalize the manuscript as much as possible in order to expand the potential audience and readers.
Some specific examples are such as: SE131; SE270 and many others local acronyms.
Author response: We would like to first thank the reviewer for the feedback. We agree that the original empirical and regional focus of the paper limits the audience. We have therefore extended the manuscript so that more emphasis is placed on methodological aspects, which make will make it more interesting to a larger group of readers. We also extended our conclusion section to compare our estimation results to similar studies on the US and China.
Specific Comments:
Abstract:
Line 13-15: Future work includes impact analysis via scenario development for temperature and precipitation similar to the work of Wang et al. (2018), yet correcting for the interdependencies between the two. Is it normal to include references in the abstract?? Please confirm this with Journal format and specifications.
Author response: We have rewritten the abstract and removed the reference.
Introduction: Climate and Agriculture is a discipline that existed even before the Agricultural Revolution. That said, there are many important references missing. Under the Climate Change context also mentioned in the manuscript, Food Security need to be considered in the discussion even thought I recognized it is not focus of your analysis.
Author response: We have rewritten the introduction and conclusion of the text so we establish the link to the indeed important topic of food security.
Page 2. Line 47. Ibid? please confirm this citation format with Journal rules.
Author response: Rewritten as suggested.
3. Model and Empirical Specification. Equations are numbered, however there is no reference in the text. Also, equation numbers are out of order.
Author response: Rewritten as suggested.
Please add a conclusion Section.
Author response: Added as suggested.
References are not following any specific format. Please confirm with Journal format. Authors Last name are cited on the text, however, in the reference section, "References" are numbered. This makes the reviewer's job difficult to verify the correct use of scientific literature.
Author response: Rewritten as suggested.
Reviewer 2 Report
Please see the attached file.

Author Response
Review of The connection Between Weather Volatility and Enterprise Efficiency as Exemplified by the European Case Authors: Angelova and Kabel
The authors make the case that agricultural output from members of the EU are directly affected by changes in weather/climate. While the paper’s goal is admirable and important, I feel they did not properly execute the methods, nor did they properly describe the results/discussion in context of the literature. I have line by line comments below, with my greatest concerns outlined in detail. At this stage, I suggest a rejection of this manuscript until the authors are able to submit as a new manuscript with a more thorough job of representing their results and their approach.
Author response: We would like to first thank the reviewer for the feedback. We have rewritten and augmented large parts of the manuscript. We feel that the comments were most helpful.
Line 61: “…climatic effects on agriculture…”
Author response: Rewritten as suggested.
Line 69: resent should be recent
Author response: Rewritten as suggested.
Line 70: Total factor productivity should be defined.
Author response: Definition added as suggested.
Line 73: What is DICE?
Author response: Definition added as suggested.
Line 71: when you say “precipitation shocks” are you meaning people are surprised by the event? This needs some additional explanation since you use it again later in the paragraph.
Author response: Explanation added as suggested.
Line 76-78: please reread. This sentence is missing a few words. Also, you need to explain sectoral disaggregation.
Author response: Rewritten, explanation added as suggested.
Line 80: “describes” should be describe
Author response: Rewritten as suggested.
Lind 81: You use “TFP” but don’t introduce it earlier. I assume it is total factor productivity, but this should be clear.
Author response: Explanation added as suggested.
Line 84: 1,5 is 1.5 (looks to be the same wherever you have a percentage. Please correct throughout).
Author response: A consistent decimal format with dots has been adopted throughout the manuscript.
Line 82: “Innovation activity”? This sentence should be cited as written. You make it sound like the U.S. has the capacity to slow our direct impacts of weather shocks because of innovations, please explain.
Author response: Explanation added as suggested.
Line 96: THI? And what is the Oury index?
Author response: Explanation added as suggested.
Section 3: There are many terms in these equations that are not being defined? What is Theta? What is v? Make sure each are defined. Trying to be simple is great, but not if it leaves off necessary information for the reader.
Author response: Explanation added as suggested.
Line 124-125: you have the word data three times in one sentence, please rephrase.
Author response: Rewritten as suggested.
Section 4: A map should be included that shows where the farms are from the 28 member states. Results are always rooted in geography and it could be an interesting way to think about your results.
Author response: Maps have been added as suggested.
Line 138: data are Line 163: Data are Line 168: data were
Author response: Rewritten as suggested.
Line 168: why are Cyprus and Malta missing?
Author response: Unofficial information from Copernicus informs us that both small islands of Cyprus and Malta are too small in terms of geography for climate scientists to generate reliable weather data through averaging between nearby weather stations. They are missing from the dataset of Copernicus all together at this point but will be introduced in the future once the technical issues are solved.
Line 170: In table one, are the output and input in millions of euros? Or billions?
Author response: Both input and output are in billions. The passage has been rewritten as suggested.
And what is pre variance? Oh - is it precipitation variance? You might change that to precip variance? Pre variance sounds like a variance before something…
Author response: The phrase has been rewritten as suggested.
Is the temperature variance really 161 degrees in the maximum? Please verify.
Author response: Yes, this was observed in Finland in the climatically extreme year 2010 with a cold winter and a very hot summer. (Reference: https://en.ilmatieteenlaitos.fi/press-release/125205)
Also, what is 1st quarter vs. 3rd quarter? This table requires far more description in the text as well as in the caption. I’ll leave it to the editor to decide the description quality in the caption - some journals require less/more. Either way, more text in the manuscript is needed.
Author response: They refer to the quartiles. An explanation has been added in the text.
Line 174: Please describe what a unit root test is.
Author response: An explanation has been added in the text.
Paragraph starting at 192: Please read through this carefully. It is missing some necessary words for explanation. What is LR?
Author response: Paragraph rewritten. LR stands for likelihood ratio, an explanation has been added in the text.
Table 3: I don’t understand the results in your sigma-squared column. Please assist the reader.
Author response: An explanation has been added in the text.
Table 4: Does a value of 1 mean they are as efficient as they always were? Or average efficiency? Or you are expecting them to be more efficient? Far more explanation is needed in the manuscript to support both Table 3 and Table 4.
Author response: It means that they are as efficient as they can be. An explanation has been added in the text.
Refer to your results in the manuscript and describe them. Pick a country and pick a couple of years. Walk the reader through interpreting your table.
Author response: An explanation has been added in the text. We have offered an interpretation of our regression output at the sample mean.
Your results and discussion section are severely lacking. You say the results are directly comparable to Wang et al 2018, but you never compare them. You are trying to make huge connections between changes in climate variability and resultant change in agricultural output, but your results are never described in the context of future production, nor in the context of comparing it to other locations around the world.
Author response: A conclusion section has been added to the manuscript. The result section has been thoroughly rewritten. The new version of the manuscript compares the results to other locations around the world and to Wang et al 2018.
You mention “nonstationary data regression methods” but do very little to describe said methods.
Author response: The text has been rewritten to lead the interested reader to the methods in question. References have been added.
Table 3 describes these regression results, but have you ever tested the assumptions of the regression models? Meaning, have you checked residual distribution to be sure the models are adequate?
Author response: The tests have been performed and the results have been reported in the text. The models are valid.
It is as though your paper just abruptly ends, so the authors need to take care of further explaining their results and placing them in the context of the literature.
Author response: A conclusion section has been added to the manuscript. The result section has been thoroughly rewritten.
Finding a way to graphically represent some of this information would be extremely useful. Again, it might help to facilitate some conversations about the geographic distribution of the results.
Author response: Maps have been added as suggested.The geographical distribution has been discussed in the text.
Reviewer 3 Report
Paper title: The connection Between Weather Volatility and Enterprise Efficiency as Exemplified by the European Case
Comments for the Authors:
You mention “enterprise” only in the title. Adjust title to the paper.
Abstract s not informative enough. For start, efficiency is a broad term. Make clear of what efficiency are you speaking. In title you say Enterprise efficiency, trough the paper I got an impression you talk about agricultural production efficiency, yet in abstract you say nothing. Remove reference and future aspires from abstract. Your abstract is so lean and under-informative, do not waste words on other people work. Focus on your current paper: name methodology used, state your key findings and scientific contribution of your paper. Introduction has greater focus on climate change that on research problem. Needs to be improved.
Introduction should provide reader an insight to the research problem and methodology used. State your hypothesis or research problems. Explain why is topic relevant and current.
Do not use (ibid) in Harvard referencing. If the source of several findings or data is repeating in subsequent sentences, just state the source once at the end of citation. If sentences are not subsequent, write the full reference again. Do not use abbreviation without giving the full term first: case of RCP, DICE, TFP.
Line 69: change “resent” into “recent”.
Use “.” Instead of “,” for decimal places.
Formulas (2) & (3) need further explanation. What do v, z, t or w stand for? It is not self-explanatory.
In section Data, you explain database design. If data is available for 25 out of 28 EU countries from 2004 to 2016, why do you use data for only 18 member states (lines 138 -144)? And in table 4 you show results for 21 state. Why do not numbers match?
Majority of paragraph containing lines 145-157 not needed because it is not relevant to research problem. Just naming six thematic issues is enough.
In Table 1 do not use abbreviation as if they are self-explainable. For example, Pre variation. Either use the full term of provide legend beneath the table.
Line 194: change “freedon” to “freedom”.
Discussion of results quite poor. Give more emphasis on table 4. Would add to paper quality if results of your study are linked (compared, contrasted) to work of other authors.
Need to add conclusion section. Give a summary of most important theoretical and empirical insights. Clearly define your scientific contribution. Provide business implications or implications for practice in general. Name research limitation, if any. Give recommendations for future research.
Author Response
You mention “enterprise” only in the title. Adjust title to the paper.
Author response: We would like to first thank the reviewer for the feedback. Rewritten as suggested.
Abstract s not informative enough. For start, efficiency is a broad term. Make clear of what efficiency are you speaking. In title you say Enterprise efficiency, trough the paper I got an impression you talk about agricultural production efficiency, yet in abstract you say nothing. Remove reference and future aspires from abstract. Your abstract is so lean and under-informative, do not waste words on other people work. Focus on your current paper: name methodology used, state your key findings and scientific contribution of your paper.
Author response: Rewritten as suggested, efficiency defined, references removed.
Introduction has greater focus on climate change that on research problem. Needs to be improved.
Author response: Rewritten as suggested.
Introduction should provide reader an insight to the research problem and methodology used. State your hypothesis or research problems. Explain why is topic relevant and current.
Author response: Rewritten as suggested.
Do not use (ibid) in Harvard referencing. If the source of several findings or data is repeating in subsequent sentences, just state the source once at the end of citation. If sentences are not subsequent, write the full reference again.
Author response: Rewritten as suggested.
Do not use abbreviation without giving the full term first: case of RCP, DICE, TFP.
Author response: Rewritten as suggested.
Line 69: change “resent” into “recent”.
Author response: Rewritten as suggested.
Use “.” Instead of “,” for decimal places.
Author response: Rewritten as suggested.
Formulas (2) & (3) need further explanation. What do v, z, t or w stand for? It is not self-explanatory.
Author response: Rewritten as suggested.
In section Data, you explain database design. If data is available for 25 out of 28 EU countries from 2004 to 2016, why do you use data for only 18 member states (lines 138 -144)? And in table 4 you show results for 21 state. Why do not numbers match?
Author response: Indeed some of the observations for the EU countries had to be dropped as there was concern regarding the corresponding weather observations. An explanation has been added in the text. The number 18 in the original manuscript was a typo, which has been corrected in the new version. The correct number is 21.
Majority of paragraph containing lines 145-157 not needed because it is not relevant to research problem. Just naming six thematic issues is enough.
Author response: Rewritten as suggested.
In Table 1 do not use abbreviation as if they are self-explainable. For example, Pre variation. Either use the full term of provide legend beneath the table.
Author response: Rewritten as suggested.
Line 194: change “freedon” to “freedom”.
Author response: Rewritten as suggested.
Discussion of results quite poor. Give more emphasis on table 4. Would add to paper quality if results of your study are linked (compared, contrasted) to work of other authors.
Author response:The result section has been thoroughly rewritten. A comparison of the results is added in conclusion section.
Need to add conclusion section. Give a summary of most important theoretical and empirical insights. Clearly define your scientific contribution. Provide business implications or implications for practice in general. Name research limitation, if any. Give recommendations for future research.
Author response: A conclusion section has been added to the manuscript which reflects all the suggested points.
Round 2
Reviewer 1 Report
After reading the new version, I could say that the authors covered all the concerns mentioned on the first revision.
Minor comments: Please confirm that Table 3 is actually a Table and not a Figure.
Is this the new Title?: Weather Volatility and Production Efficiency. Please
Page 1. Line 41: the word "wafrm" should be "warm"
Author Response
Reviewer: After reading the new version, I could say that the authors covered all the concerns mentioned on the first revision.
Minor comments: Please confirm that Table 3 is actually a Table and not a Figure.
Authors: Thank you. Table 3 is indeed a Figure. Changes to this effect have been made in the text.
Reviewer: Is this the new Title?: Weather Volatility and Production Efficiency. Please
Authors: Yes, this is the new title.
Reviewer: Page 1. Line 41: the word "wafrm" should be "warm"
Authors: Corrected as suggested.
Reviewer 2 Report
The authors have done a nice job fixing the problems the reviewers had originally suggested. I recommend publishing.
Author Response
Reviewer: The authors have done a nice job fixing the problems the reviewers had originally suggested. I recommend publishing.
Authors: Thank you.